# WHY DO WE NEED WEIGHT DECAY IN MODERN DEEP LEARNING?

## ABSTRACT

Weight decay is a broadly used technique for training state-of-the-art deep networks, including large language models. Despite its widespread usage, its role remains poorly understood. In this work, we highlight that the role of weight decay in modern deep learning is different from its regularization effect studied in classical learning theory. For overparameterized deep networks, we show how weight decay modifies the optimization dynamics enhancing the ever-present implicit regularization of SGD via the *loss stabilization mechanism*. In contrast, for large language models trained with nearly online SGD, we describe how weight decay balances the *bias-variance tradeoff* in stochastic optimization leading to lower training loss. Moreover, we show that weight decay also prevents sudden loss divergences for `bfloat16` mixed-precision training which is a crucial tool for LLM training. Overall, we present a unifying perspective from ResNets on vision tasks to LLMs: weight decay is never useful as an explicit regularizer but instead changes the training dynamics in a desirable way.

## 1 INTRODUCTION

Weight decay and $\ell_2$ regularization are widely studied topics in machine learning. Weight decay serves to constrain the network capacity (Goodfellow et al., 2016) and acts as a mechanism for suppressing irrelevant weight components, aligning with the principles of Occam's razor (Krogh & Hertz, 1991). It is central in discussions on generalization bounds (Shalev-Shwartz & Ben-David, 2014), albeit a recent empirical study by Jiang et al. (2020) casts doubt on how well norm-based measures correlate with generalization for deep networks. Weight decay is also known to yield a regularization of the input-output Jacobian (Zhang et al., 2018) and to alter the training dynamics of scale-invariant networks by changing the *effective* learning rate (Van Laarhoven, 2017).

**Why revisiting weight decay now?** Weight decay is widely used for training *most state-of-the-art* deep networks such as GPT-3 (Brown et al., 2020), CLIP (Radford et al., 2021), or PALM (Chowdhery et al., 2022). We argue that despite its widespread usage, its effect is still poorly understood: in some cases it acts as a regularizer but in some cases as a tool for better optimization. Although the regularization effect of weight decay is thoroughly studied in classical learning theory, deep networks are already equipped with strong *implicit* regularization coming from the parameter initialization, optimization algorithm, and architecture (Zhang et al., 2016). Moreover, recent years have brought along new architectures and settings such as transformers (Vaswani et al., 2017) and nearly one-epoch language modelling (Brown et al., 2020; Hoffmann et al., 2022). All of this makes it unclear to what extent classical results are applicable to modern deep learning settings.

**Contributions.** Our work aims to provide a systematic answer to the following question:

<p align="center">Why do we need weight decay in modern deep learning?</p>

Towards this goal, we make the following contributions:

- For overparameterized networks, we provide a unifying view on the mechanism by which weight decay enhances the implicit regularization effect of the SGD noise. We show that the trajectory stays close to the trajectory of a process where the trace of the Hessian is regularized. This analysis unveils the role of learning rate and weight decay on generalization.
- In contrast, for LLMs trained with nearly one-pass SGD, weight decay does not show an important regularization effect. Instead, we reveal how it modifies the effective learning rate and

better balances the *bias-variance optimization tradeoff* leading to lower training loss. We discuss practical implications related to this tradeoff.

- Moreover, we show that weight decay also prevents sudden loss divergences for `bfloat16` mixed-precision training which is a crucial tool for LLM training at scale.

We conclude that weight decay is rarely useful as an explicit regularizer but instead its wide usage can be attributed to its ability to change the optimization dynamics in a desirable way.

## 2 RELATED WORK

We discuss the most related works here and provide further comparisons later in the paper.

The concept of employing $\ell_2$ weight penalty traces back to studies on the stability of solutions for ill-posed problems (Tikhonov, 1943). It has since been extensively explored in statistics (Foster, 1961; Hoerl, 1962; Hoerl & Kennard, 1970). Krogh & Hertz (1991) present one of the earliest systematic studies on weight decay tailored for *neural networks*. Generalization bounds, such as those by Shalev-Shwartz & Ben-David (2014), suggest that weight decay can be *sufficient* for generalization, although not strictly necessary, e.g., due to the implicit regularization of gradient methods (Soudry et al., 2018). Zhang et al. (2016) argue that while weight decay improves test accuracy, the improvement is not substantial ($\approx$ 1-2% on ImageNet), indicating the key role of implicit regularization. Loshchilov & Hutter (2019) highlight the distinct effects of weight decay and $\ell_2$ regularization, particularly for Adam, suggesting that weight decay leads to superior regularization and simpler hyperparameter tuning. For GPT-3 training, Brown et al. (2020) suggest that they include weight decay to provide *a small amount of regularization*, although we believe it is not the primary reason as we discuss in Sec. 4.

Multiple works have focused on weight decay as a tool influencing optimization dynamics. Van Laarhoven (2017) emphasizes that weight decay's impact on scale-invariant networks is primarily seen in terms of an effective learning rate. Zhang et al. (2018) propose three mechanisms of weight decay regularization: (1) increasing the effective learning rate for scale-invariant networks, although as we discuss, the same holds even for networks *without any normalization layers*, (2) approximating the regularization of the input Jacobian for an optimizer inspired by second-order methods, (3) inducing a specific dampening effect in this optimizer. Li & Arora (2019); Li et al. (2020) explore the optimization properties of scale-invariant deep networks for which the effective learning rate can be formally derived. Lewkowycz & Gur-Ari (2020) suggest that the best generalization is achieved with the smallest $\lambda_{WD}$ although it necessitates longer training. Additionally, Lewkowycz (2021) propose a criterion for detecting when to decay the learning rate based on the evolution of the weight norm. Lastly, Li et al. (2022a) make the BERT architecture scale-invariant to enhance training stability and make it more compatible with standard SGD.

The seminal paper of Krizhevsky et al. (2012) that introduced AlexNet suggest that weight decay serves not only as a regularizer but also reduces the model's training error, functioning as an *optimization tool*. In recent work, Hoffmann et al. (2022) briefly observe that weight decay enhances the training performance of Adam for training LLMs, but only after $\approx 80\%$ of the total iterations. However, they do not provide an explanation for this behavior, a point we delve into in Sec. 4.

## 3 WEIGHT DECAY FOR OVERPARAMETERIZED DEEP NETWORKS

In this section, we delve into the influence of weight decay in overparameterized settings, with a specific focus on image classification tasks. We first examine its impact on training VGG and ResNet models using SGD on CIFAR-10 and CIFAR-100 datasets. Then, the analysis of a simplified setup provides foundational insights, elucidating the role of weight decay in broader training scenarios.

**Notations and setup.** Let $(x_i, y_i)_{i=1}^n$ be the training inputs and labels where $x_i \in \mathcal{D}$, $y_i \in \mathbb{R}^c$, and $c$ is number of classes. Let $h : \mathbb{R}^p \times \mathcal{D} \to \mathbb{R}^c$ be the hypothesis class of neural network and for any parameter $\mathbf{w} \in \mathbb{R}^p$ where the function $h(\mathbf{w}, \cdot) : \mathcal{D} \to \mathbb{R}^c$ represents the network predictions. We assume for this section that the network is overparameterized and capable of achieving perfect

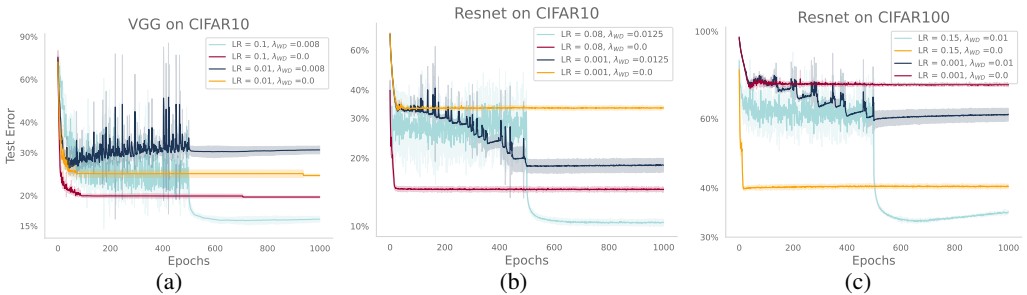

Figure 1: **Training with and w/o weight decay.** We report the test error for VGG (1a) and ResNet (1b, 1c) trained on CIFAR-10/100 with and without weight decay and with small and large learning rates. After the first 500 epochs the learning rate is decayed to $\eta = 10^{-4}$ for all the curves.

training accuracy. The training loss $\mathcal{L}$ and the $\ell_2$-regularized training loss $\mathcal{L}_\lambda$ are given as:

$$\mathcal{L}(\mathbf{w}) := \frac{1}{N} \sum_{i=1}^{N} \ell\left(y_i, h(\mathbf{w}, x_i)\right), \quad \mathcal{L}_\lambda(\mathbf{w}) := \mathcal{L}(\mathbf{w}) + \frac{\lambda}{2} \|\mathbf{w}\|^2, \quad (1)$$

where $\ell(\cdot, \cdot) : \mathbb{R}^c \times \mathbb{R}^c \to \mathbb{R}$ denotes the cross-entropy loss function. With $i_t \sim \mathbb{U}([N])$, the SGD algorithm on $\mathcal{L}_\lambda(\mathbf{w})$ (here with batch size 1 and with replacement) with a learning rate (LR) $\eta$ is

$$\mathbf{w}_{t+1} = \mathbf{w}_t - \eta \nabla_{\mathbf{w}} \ell\left(y_{i_t}, h(\mathbf{w}_t, x_{i_t})\right) - \eta \lambda \mathbf{w}_t. \quad (2)$$

**Experimental setup.** We train VGG (Simonyan & Zisserman, 2014) without BatchNorm and ResNet (He et al., 2016) models on CIFAR-10/CIFAR-100 using SGD and step-decay (He et al., 2016) as LR schedule. Moreover, we compare different values of $\ell_2$-regularization coefficient $\lambda$. By decaying the LR we divide the training into two separate phases: (1) **large-LR phase** which uses a large constant LR to exploit the SGD noise, and (2) **fine-tuning phase** which uses a small LR to converge to a minimum of the problem.

### 3.1 DIFFERENT MECHANISMS OF TRAINING WITH WEIGHT DECAY

To understand whether minimizing the regularized objective in Eq. (1) alone ensures optimal generalization, we compare test errors in Fig. 1 across various settings with large and small LR. The necessity of a high LR for optimal performance suggests that optimizing the regularized objective is insufficient to explain the benefits of WD—*the regularized objective alone does not guarantee generalization*. This experiment reaffirms the widely acknowledged consensus that *implicit regularization induced by the LR is crucial* (Keskar et al., 2016; Li et al., 2019; Andriushchenko et al., 2023). Despite revealing an interplay between weight decay and large initial LR, the understanding of the corresponding dynamics remains limited. In this section, our goal is to comprehensively understand these dynamics, particularly to elucidate the distinctions between the yellow and turquoise curves in Fig. 1 and the resulting differences in their generalization.

Given the regularization of the $\ell_2$ norm of parameters, it is natural to wonder whether weight decay's improvement primarily stems from its ability to control the norm of the trained model. The experiment in Fig. 3a clearly illustrates that distinct training trajectories, while resulting in the same final $\ell_2$ norm for parameters, can yield different levels of generalization stating that *the $\ell_2$-norm of the learned model's parameters is inconsequential*. This observation suggests that once the norm is constrained by weight decay, the critical factor influencing the model's generalization is the subsequent choice of LR. Note that predictions can be scale-independent for various reasons (such as normalization layers or homogeneous activation), making the parameters' scale inconsequential.

Understanding the impact of WD on the optimization dynamics is crucial for grasping its benefits in generalization. We start by examining the parameter norm evolution in Fig. 3a. It rapidly decays to stabilize within a small, approximately constant interval. After the rapid decrease, the optimization resembles the dynamics of SGD projected onto a sphere with a certain radius. We assert that this stage is pivotal for training with weight decay and hypothesize the following key mechanism:

*Weight decay maintains parameters norm in a small bounded interval. The resulting projected noise-driven process induces an implicit regularization effect.*

The rest of the section is dedicated to empirically confirming this implicit regularization mechanism.

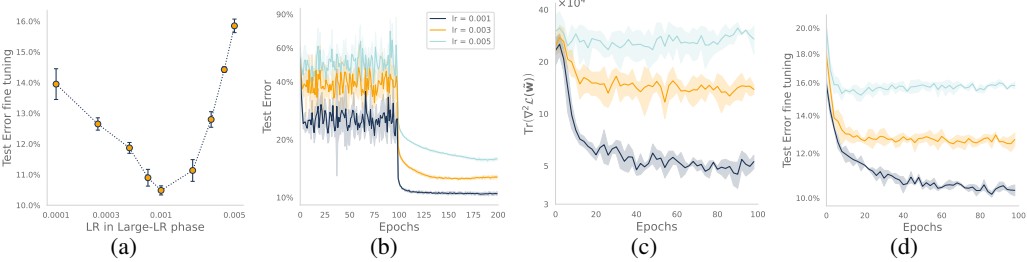

Figure 2: **Training scale-invariant ResNets on the sphere.** We train on CIFAR-10 with three different large LR for the first 100 epochs and decay it to $\eta = 10^{-4}$ afterwards. Figure (2a) reports the test error with respect to different LRs in the first phase showing the existence of an optimal value. Figure (2d) reports the test error along the iterations. Figures (2c, 2d) report the decreasing trend of the trace of the Hessian and test error after fine-tuning for 100 epochs with $\eta = 10^{-4}$ every 2 epochs.

## 3.2 WARMUP: OPTIMIZATION ON THE SPHERE WITH SCALE INVARIANCE

In order to isolate the implicit regularization mechanism from the large initial drop of the $\ell_2$ norm, we consider a simplified setting. We train scale-invariant networks (Li & Arora, 2019; Li et al., 2020) with projected SGD on the unitary sphere $\mathbb{S}^{(p-1)}$. This setup is advantageous for two reasons: (a) it streamlines LR selection, significantly reducing experimental complexity, and (b) prior research on scale-invariant networks (Li & Arora, 2019; Li et al., 2020; Kodryan et al., 2022) facilitates a clear comparison to our result. The projected SGD update writes as

$$\mathbf{w}_{t+1} = \Pi_{\mathbb{S}^{(p-1)}} \left( \mathbf{w}_t - \eta \nabla_{\mathbf{w}} \ell \left( y_{i_t}, h(\mathbf{w}_t, x_{i_t}) \right) \right) \quad \text{where} \quad \Pi_{\mathbb{S}^{(p-1)}} : \mathbf{w} \mapsto \mathbf{w}/\left\| \mathbf{w} \right\|_2. \tag{3}$$

The training framework still consists of two phases separated by a LR decay. The primary insight from our experiments on the sphere is depicted in Fig. 2: the test performance achieved in the fine-tuning phase depends on the LR used in the large-LR phase and, moreover, there is an optimal value. Our investigation reveals that the key to understand this behavior and the dependence on the LR lies in the noisy dynamics in the first phase.

**The noise driven process.** We introduce the key ingredients of SGD noise and subsequently exploit the properties of their approximations to investigate the implicit regularization effect. Let $g_t = \nabla_{\mathbf{w}} \mathcal{L}(\mathbf{w}_t) - \nabla_{\mathbf{w}} \ell(y_{i_t}, h(\mathbf{w}_t, x_{i_t}))$ denote the noise in the gradient.

(**P1**) Under reasonable approximations (details in Prop. 3) the scale of the noise is proportional to the train cross-entropy loss, i.e., $\mathbb{E}\left[ \|g_t\|^2 \right] \sim \mathcal{L}(\mathbf{w}_t)$. Hence, a higher training loss implies a larger noise in the stochastic gradients. The experiments in Fig. 11, 12 show that in the large LR phase, the training loss remains nearly constant. Based on this observation, we assume $\mathbb{E}\left[ \|g_t\|^2 \right] \asymp \sigma_\eta^2$.

(**P2**) We empirically observe that the covariance of the noise $\Sigma_t = \mathbb{E}\left[ g_t g_t^\top \right]$ and the Hessian $\nabla_{\mathbf{w}}^2 \mathcal{L}(\mathbf{w}_t)$ have the same shape, see App. C.4.

In the case of regression, the shape of the covariance of the stochastic gradients, when the labels are injected with Gaussian noise, also matches the shape of the Hessian. This crucial observation is used in several works (Blanc et al., 2020; Li et al., 2021; Damian et al., 2021) to demonstrate the implicit regularization properties of SGD. Specifically, Damian et al. (2021); Pillaud-Vivien et al. (2022) show that the SGD trajectory closely tracks the solution of a regularized problem. Leveraging property (**P2**), we conjecture that a similar result should hold in our analysis and that the dynamics of SGD on the sphere for classification tracks closely a regularized process.

**Conjecture 1.** *Consider the algorithm Eq. 3 with $\mathbf{w}_0$ initialized from a distribution $\mu_0 \left( \mathbb{S}^{(p-1)} \right)$. For any input $x$, let $\mathbf{w}_t, h(\mathbf{w}_t, x)$ be the random variables that denote the iterate at time $t$ and its functional value. The stochastic process $(h(\mathbf{w}_t, x))_{t \in \mathbb{N}}$ will converge to a stationary distribution $\mu_\eta^\infty(x)$ with mean $\bar{\mu}_\eta(x)$ for which the following property holds,*

$$\bar{\mu}_\eta(x) = h\left( \mathbf{w}_\eta^*, x \right), \quad \text{where} \quad \mathbf{w}_\eta^* := \underset{\mathbf{w} \in \mathbb{S}^{(p-1)}}{\arg\min} \ \mathcal{L}(\mathbf{w}) + \eta \sigma_\eta^2 \operatorname{Tr}\left( \nabla^2 \mathcal{L}(\mathbf{w}) \right). \tag{4}$$

The important difference in our statement is that, unlike Blanc et al. (2020); Damian et al. (2021), we do not need to add noise to the labels at each iteration. Instead, the large-LR phase induces a label noise-like behavior similar to Andriushchenko et al. (2023).

**Mixing in the function space.** A simpler conjecture could have been that the iterates $(\mathbf{w}_t)_{t\geq 0}$ mix towards a solution of the regularized objective $\mathbf{w}_\eta^*$. However, Li et al. (2020) argues against mixing in the parameter space, emphasizing the necessity of considering the function space. Hence, our conjecture is formulated to capture stationarity in function space.

**What is the purpose of the fine-tuning phase?** Even at stationarity,[1] the values of the loss $\mathcal{L}(\mathbf{w}_t)$ and of $\mathrm{Tr}\left(\nabla^2\mathcal{L}(\mathbf{w}_t)\right)$ are still dominated by the noise. This noise obscures any discernible trend along the trajectory, making it challenging to argue convincingly about convergence to the minimum of the regularized loss. While Langevin dynamics suggest LR annealing to approach the mean of the stationary distribution, this technique does not fully resolve the issue. The noise is state-dependent and decreasing the LR might change the stationary distribution and potentially the regularized objective. An alternative approach is to project the iterate $\mathbf{w}_t$ onto a manifold where the loss matches the value evaluated at the mean. Analyzing the evolution of $\mathrm{Tr}\left(\nabla^2\mathcal{L}\right)$ at these projected iterates might reveal evidence of a regularized process. For illustrative image, refer to Fig. 10. This projection corresponds to the fine-tuning phase and is accomplished with early-stopped gradient flow (SGD with a small LR).

**Interpretation of the conjecture and links to generalization.** The empirical observations in Fig. 2 show that when two different LRs $\eta_l$ (large) and $\eta_s$ (small) are used in the large-LR phase, models with different generalization properties are obtained after the fine-tuning phase. Our conjecture explains this gap as two solutions $\bar{\mu}_{\eta_l}$ and $\bar{\mu}_{\eta_s}$ of the regularized problem having different strength of regularization ($\eta_l\sigma_{\eta_l}^2$ vs $\eta_s\sigma_{\eta_s}^2$). The solution $\bar{\mu}_{\eta_l}$ benefits from better regularization and therefore endows better generalization properties. The conjecture further explains the U-shape generalization curve in Fig. 2a where optimal regularization results in good test performance, and models beyond that level are over-regularized. The regularization is implicit and is solely due to the noisy dynamics.

**Revealing the implicit regularization mechanism.** Here, we present additional empirical evidence that the process $(\mathbf{w}_t)_{t>0}$ closely tracks the regularized process. As mentioned earlier, directly measuring the loss $\mathcal{L}$ or $\mathrm{Tr}\left(\nabla^2\mathcal{L}\right)$ at $\mathbf{w}_t$ fails to reveal any decreasing trend due to noise interference. Therefore, we utilize the fine-tuning process to exhibit this decreasing trend. During fine-tuning, the iterate $\mathbf{w}_t$ is projected to a nearby point, denoted as $\tilde{\mathbf{w}}_t$, such that $\mathcal{L}(\tilde{\mathbf{w}}_t) \sim \mathcal{L}(\mathbf{w}_\eta^*)$. Since their loss values are similar, we compare $\mathrm{Tr}\left(\nabla^2\mathcal{L}(.)\right)$ at $\mathbf{w}_\eta^*$ and $\tilde{\mathbf{w}}_t$. In the experiments detailed in Fig. 2c, we report $\mathrm{Tr}\left(\nabla^2\mathcal{L}(.)\right)$ along the fine-tuned iterates $\tilde{\mathbf{w}}_t$ and observe a decreasing trend. The trajectory of the iterates $(\mathbf{w}_t)_{t\geq 0}$ closely follows the trajectory of the fine-tuned iterates $(\tilde{\mathbf{w}}_t)_{t\geq 0}$ which converge to $\mathbf{w}_\eta^*$. This mechanism elucidates how the trajectory of SGD implicitly biases the model towards a regularized solution that leads to enhanced generalization properties.

**Comparison with the related works.** Our focus is to empirically illustrate the implicit regularization phenomenon, and we refrain from attempting to prove this general conjecture, which we consider a challenging task. We refer to App. C.3 for comparison with further works related to label noise. Li et al. (2020) also studies the stochastic process that governs the evolution of parameter directions in scale-invariant networks. Nevertheless, our approach differs in nature as we aim to provide a qualitative description of the stationary distribution (see App. C.3 for more details).

### 3.3 A UNIFYING THEME: BEYOND SCALE INVARIANCE AND SPHERICAL OPTIMIZATION

The spherical case studied in the previous subsection paints a clear picture. When isolated from the evolution of the norm, the stochastic dynamics induced by SGD and large LRs provide better control over the trace of the Hessian of the model and thus enforce a useful regularization which translates into good generalization properties. In this section, we demonstrate that a similar picture holds in the case of standard training with weight decay. We extend the Conjecture 1, to hold beyond spherical optimization and for networks which are not scale invariant.

**Conjecture 2.** *Consider the algorithm in Eq. 2 with $\mathbf{w}_0$ initialized from a distribution $\mu_0\left(\mathbb{R}^{(p)}\right)$. For any input $x$, let $\mathbf{w}_t, h(\mathbf{w}_t, x)$ be the random variables that denote the iterate at time $t$ and its functional value. The stochastic process $(h(\mathbf{w}_t, x))_{t\in\mathbb{N}}$ converges to the stationary distribution*

---

[1]Assuming the existence of a stationary distribution, the iterates $\mathbf{w}_t$ are eventually realizations from this distribution.

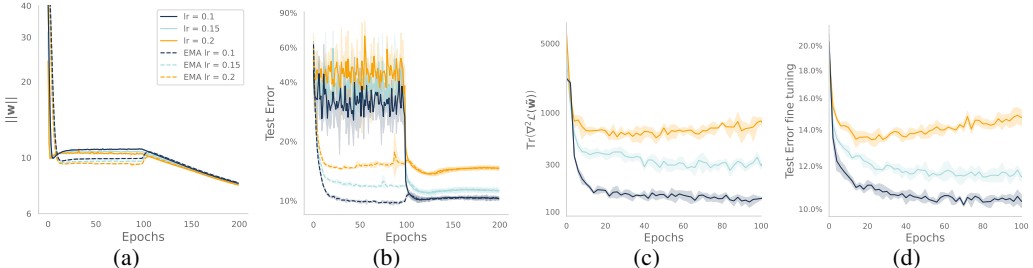

Figure 3: **Training standard ResNets with weight decay.** We train on CIFAR-10 with $\lambda_{WD} = 0.015$, three different large LRs for the first 100 epochs and decay them to $\eta = 10^{-3}$ afterwards. The norm in Fig. 3a converges to the same value after the LR decay while the test error in Fig. 3b is different. Fig. (3c, 3d) report the decreasing trend of $\mathrm{Tr}(\nabla^2)$ and test error after fine-tuning for 100 epochs with $\eta = 10^{-3}$ every 2 epochs.

$\mu_{\eta,\lambda}^{\infty}(x)$ *with mean* $\bar{\mu}_{\eta,\lambda}(x)$ *for which the following property holds,*

$$\bar{\mu}_{\eta,\lambda}(x) = h\left(\mathbf{w}_{\eta,\lambda}^{*}, x\right), \text{ where } \mathbf{w}_{\eta,\lambda}^{*} := \underset{\mathbf{w} \in \mathbb{R}^p}{\arg\min} \ \mathcal{L}_{\lambda}(\mathbf{w}) + \eta \sigma_{\eta,\lambda}^2 \ \mathrm{Tr}\left(\nabla^2 \mathcal{L}(\mathbf{w})\right). \quad (5)$$

There are two differences compared to Conjecture 1: (a) the loss term in the regularized objective is replaced by a $\ell_2$-regularized loss and (b) most importantly the strength of the regularization $\sigma_{\eta,\lambda}$, now depends on both the LR and the WD parameter $\lambda$. Our experiments in Fig. 3, provide empirical validation for this conjecture. When trained with different LRs and then fine-tuned, the training converges to models with different test performances. This difference is primarily attributed to the varying regularization strengths $\sigma_{\eta,\lambda}$. The model with the largest LR exhibits the smallest $\mathrm{Tr}(\nabla^2)$. When fine-tuning every two epochs along the trajectory as reported in Fig. 3c, the quantity $\mathrm{Tr}(\nabla^2)$ is decreasing closely following a regularized process. A similar trend can be observed in Fig. 3d for the test performance when fine-tuning along the trajectory. These observations strongly indicate the benefits of generalization arising from implicit regularization.

**Exponential moving average.** As discussed in the spherical case, the iterates are noisy realizations and measuring either $\mathcal{L}$ or $\mathrm{Tr}(\nabla^2 \mathcal{L})$ at the iterates is not informative. However, we can reduce the noise by averaging and unlike the spherical case it is easy to compute the iterate average in the unconstrained case.[2] Intuitively the average should be close to $\mathbf{w}_{\eta,\lambda}^{*}$, the experiment in Fig. 3b confirms this intuition. We consider an exponential moving average (EMA) of the SGD iterates with parameter $\beta = 0.999$ and show that the test error is lower for a large LR (0.1) which enjoy better regularization. This provides further justification for our conjecture and also highlights the practical advantage of obtaining the best model by a simple exponential moving average instead of fine-tuning.

**Effective learning rate vs. high training loss.** Existing works (Zhang et al., 2018) have explored the relationship between LR and WD, introducing the concept of effective LR. These works primarily emphasize that training with WD results in a higher effective LR, without clarifying how this high LR contributes to improved generalization. We address this gap by proposing that a higher LR leads to an increase in $\sigma_{\eta,\lambda}$, consequently enhancing generalization. We claim that examining the high training loss, which approximates the scale of $\sigma_{\eta,\lambda}$, offers a more insightful explanation for the enhanced generalization ability. Analyzing the training curve in Figure 11, the training loss remains consistently high during training with weight decay, entering a phase termed "loss stabilization," by Andriushchenko et al. (2023). We assert that WD contributes to achieving this loss stabilization phase in classification tasks, leveraging the implicit regularization induced by stochastic dynamics.

**On the benefit of normalization.** Our conjecture characterizes the mixing distribution but does not delve into the speed of the mixing process. In our experiments, we observe that normalization plays a pivotal role in the speed of mixing. Li et al. (2020) observes a similar phenomenon in the case of scale-invariant networks, specifically the fast equilibrium conjecture, which is addressed by Li et al. (2022b). We note that this phenomenon persists even when the models are not scale-invariant.

**Conclusion.** The key quantity $\sigma_{\eta,\lambda}$ governs the effective regularization strength. The primary influence of WD and a large LR lies in maintaining it at an appropriate scale. Alternative methods, such as injected label noise, dropout, or data augmentation, can also achieve this objective.

---

[2]Note that on the sphere, we need to compute the mean on a manifold which is a harder problem

## 4 WEIGHT DECAY FOR LARGE LANGUAGE MODELS

In this section, we discuss how weight decay leads to better optimization properties for training language models: it leads to lower training loss and prevents sudden loss divergences.

**Experimental setting.** We use the `NanoGPT` repository (Karpathy, 2023) for training GPT-2 models (Radford et al., 2019) on OpenWebText. We train a 124M parameter model known as GPT-2-Small for $50\,000$ iterations. For most experiments, we reduce the default context length from 1024 to 256 to ensure practicality within an academic budget. Unless mentioned otherwise we train with AdamW using batch size 256, default LR 0.0006, a short 400-iteration LR warmup, and $10\times$ cosine LR decay. We keep all other hyperparameters at their default values, see Sec B for more details.

**Weight decay in LLMs is *not* regularization.** Weight decay is a common component in training state-of-the-art LLMs like GPT-3 (Brown et al., 2020), Chinchilla (Hoffmann et al., 2022), and Llama (Touvron et al., 2023). These works consistently employ $\lambda_{WD} = 0.1$, and typically do not penalize LayerNorms, resulting in a change to the minimizer but not the minimum value. While Brown et al. (2020) suggest that weight decay offers "*a small amount of regularization*," its necessity remains unclear in the context of one-pass SGD where the population loss is directly minimized. As shown in Fig. 4, where we plot the generalization gap for three training runs, the training and validation losses remain closely aligned across different weight decay values.

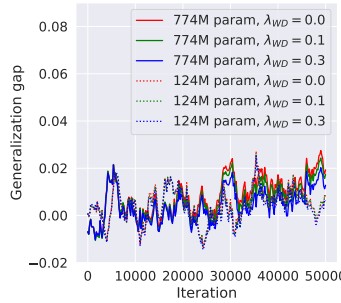

Figure 4: The generalization gap stays close to zero throughout training for different $\lambda_{WD}$.

**Two mechanisms of weight decay for LLMs.** We suggest that the two most crucial mechanisms of weight decay for LLMs are as follows: (1) better optimization as observed in Hoffmann et al. (2022), (2) prevention of loss divergences when using `bfloat16`. At first glance, adding an extra term to the optimization objective may seem counter-intuitive when the sole focus is minimizing cross-entropy loss. We offer a detailed understanding of both mechanisms that stand in contrast to the data-limited setting of Sec. 3, where optimization speed and training stability are not the primary concerns, unlike generalization.

### 4.1 UNDERSTANDING THE BENEFIT OF WEIGHT DECAY FOR BETTER OPTIMIZATION

**Better optimization is reproducible at a smaller scale.** The findings from Hoffmann et al. (2022) (Fig. A7 therein) indicate that weight decay in AdamW leads to lower training loss ($\approx 0.02$ lower), primarily towards the end of training, which also translates in a better downstream performance. We are able to reproduce this phenomenon at a smaller scale with 124M parameters in Fig. 5 (*left*): the final training loss is smaller for $\lambda$ equal to 0.1 and 0.3 compared to 0. In contrast, weight decay does not provide benefits when training with constant LRs as shown in Fig. 5 (*right*) emphasizing the importance of its interaction with LR decay. Furthermore, we observe a similarity in loss stabilization between constant LRs and weight decay. However, this stabilization does not offer utility in this context where our primary goal is improved optimization. Additionally, performing fine-tuning with a tiny LR reveals that a higher starting training loss can still be a better starting point in terms of the final loss. Moreover, in Fig. 15 in the Appendix, we show that decoupled weight decay, as advocated in Loshchilov & Hutter (2019), is unnecessary: a simple $\ell_2$ penalty added to the loss achieves the same effect. Lastly, in Fig. 16, we show that a similar improvement in training loss is also observed for *SGD with momentum* suggesting that adaptive LRs are not key for this phenomenon.

**Bias-variance tradeoff from stochastic approximation.** Stochastic approximation (SA) algorithms' convergence primarily depends on two factors: the convergence of the *bias* term and the *variance* term (Moulines & Bach, 2011). The bias term influences the rate at which initial conditions are forgotten, while the variance term results from noise in the gradient estimates. Consider the simple case of SGD with a constant LR $\eta$ applied to linear models. In this case, the expected excess risk after $t$ iterations can be bounded as

$$\text{Excess Risk} \lesssim (1 - \eta\mu)^t \|x_0 - x_*\|^2 + \eta\sigma^2,$$

where $\sigma$ is a uniform bound on the variance of the noise of gradient estimates, $\mu$ a lower bound on the objective function's Hessian, $x_0$ the initial point and $x_*$ the optimum. In the context of SA with linear models, it is well-established that a larger LR accelerates the contraction of the bias term

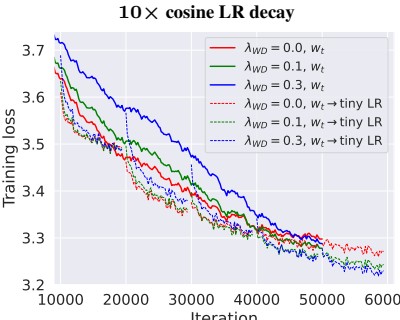 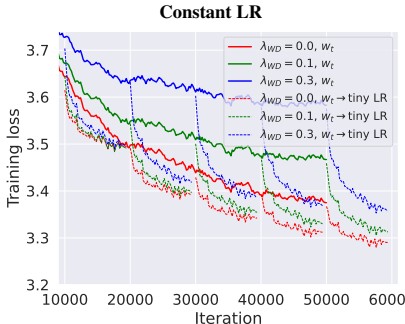

Figure 5: **GPT-2-124M on OpenWebText. Left**: We reproduce the improvement from weight decay as in Hoffmann et al. (2022) using $10\times$ cosine LR decay. Performing fine-tuning with a tiny LR reveals that a higher starting training loss can still be a better starting point in terms of the final loss. **Right**: In contrast, weight decay has no beneficial value when training with constant LRs. We see resemblance of loss stabilization which is, however, not useful in this setting.

but has a detrimental impact on the variance term. With constant LRs, the variance term eventually becomes dominant. To reduce the variance, various techniques like averaging or LR decay can be employed. However, with decaying LRs reduce the variance but simultaneously slows down the convergence of the bias term. In contrast, when employing decaying LRs, the rate of convergence can be primarily influenced by the bias term.

**Effective LR induced by weight decay.** Our main hypothesis posits that the use of WD during the training of LLM results in an increased effective LR by controlling parameter norms, even in the absence of homogeneity in the training loss. This assertion is grounded in the observed inverse correlation between the evolution of gradients and parameter norms (see Fig. 17 in the Appendix). In alignment with results for GD and scale-invariant functions, we show in Sec. D.1 in the Appendix that WD in combination with Sign GD (utilized as a surrogate for Adam) is equivalent to projected GD on the sphere, with an effective LR $\eta_{\text{eff}} \propto \eta_t / \|w_t\|_2$ (see Fig. 6). Thus, controlling parameter norms with WD allows implicit changes to the LR schedule.

**Understanding the effect of weight decay from the lens of bias-variance tradeoff.** We postulate that the effective dynamics induced by weight decay are equivalent to those without weight decay but with a higher LR. Under a constant LR, at convergence, the loss is eventually dominated by the variance and scales as $O(\eta)$. Consequently, since weight decay is equivalent to a higher LR, the final loss scales as $O(\eta_{\text{eff}})$, resulting in higher error rates, as confirmed by Fig. 5. However, with decaying LRs the picture is different. With a large LR, the convergence is still primarily influenced by the variance term, leading to higher loss values in the presence of weight decay. Conversely, in the phase with smaller LRs, bias contraction

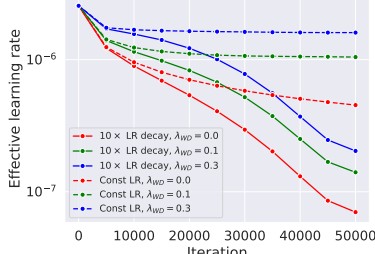

Figure 6: The effective LR for the models reported in Fig. 5.

takes precedence in the convergence process, causing weight decay to catch up and perform better at the end (see Fig. 5, *left*), thanks to its relatively higher effective LR and improved bias contraction. To support our hypothesis about effective LRs, in Fig. 19 we consider a run without weight decay but with a cosine schedule that achieves a slightly higher LR towards the end. This approach aligns with the convergence pattern observed with weight decay in the final phase, indicating that similar effects of weight decay can be replicated using a higher LR.

**Practical takeaways.** According to Sanyal et al. (2023), weight averaging for LLMs is most advantageous when employed with large LRs, underscoring the importance of reducing gradient noise in LLM training. Thus, weight averaging can serve as a nearly zero-cost proxy of fine-tuning with tiny LRs providing insight into *whether a training run is constrained by the variance term*. We show the result of weight averaging in Fig. 18 in Appendix which can be compared to Fig. 5. Interestingly, Fig. 18 also illustrates that employing a *constant* LR in conjunction with weight averaging is nearly as effective as implementing LR decay.

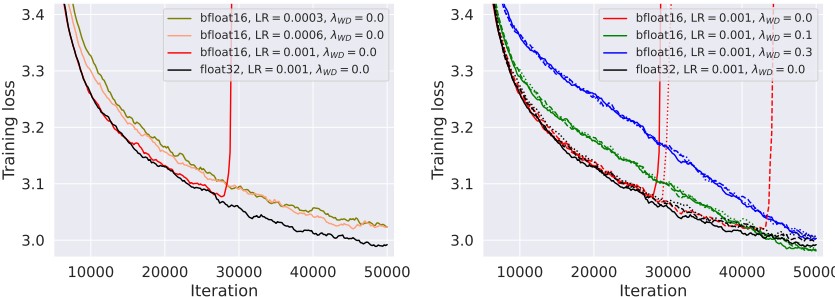

Figure 7: **GPT-2-124M on OpenWebText with context length 1024. Left:** The model trained with a moderate LR 0.001 diverges for `bfloat16` but not for `float32`; lowering the LR prevents the divergence but leads to a worse loss. **Right:** Weight decay prevents divergence for LR= 0.001 and enables `bfloat16` training (the three random seeds are denoted with —, - - -, · · · lines).

## 4.2 WEIGHT DECAY PREVENTS DIVERGENCES WITH BFLOAT16

**Overview.** Another crucial effect of weight decay is that it enables stable `bfloat16` mixed-precision training. Using `bfloat16` training helps to significantly speed up training and reduce the GPU memory requirements allowing to train larger models and use larger batches (Kalamkar et al., 2019). Scao et al. (2022) briefly observed that usage of `float16` causes spikes, while `bfloat16` is more stable. Although `bfloat16` shares the same floating-point exponent size as `float32`, it offers lower precision, with only 7 bits for the fraction instead of 23. Interestingly, we observe that even the presumably more stable `bfloat16` can still exhibit late-training spikes that irreparably harm model performance. We suspect that LLM practitioners may be aware of this phenomenon qualitatively, but we could not find any systematic reference addressing it.

**Experiments.** We observe that using a larger context length (e.g., 1024 instead of 256 as previously) makes the training more susceptible to loss divergences. Therefore, we focus on this configuration for the experiments shown in Fig. 7. We notice that runs with a moderate LR 0.001 (the default LR of Adam in `PyTorch`) without weight decay exhibit late-training divergence for *all* random seeds when using `bfloat16`, in contrast to `float32`, which remains entirely stable. Importantly, we observe that the model *does not recover* after the loss spikes, in contrast with the loss spikes described in the Edge of Stability phenomenon (Cohen et al., 2021; 2022). We note that one can prevent divergences by simply lowering the LR, e.g., from 0.001 to 0.0006 (see Fig. 7, *left*) but this leads to slower training. Instead, the most effective approach is to use a higher LR of 0.001 with weight decay, which enables stable `bfloat16` training and yields a better final training loss.

**Reasons behind the divergences.** The significance of weight decay becomes evident when we consider `float16` training, which allows only 5 bits for the exponent. Divergences in such cases are well-documented, as noted in Karamcheti et al. (2021): when moderately large values exceeding 65 519 are encountered during a *forward* pass, they are interpreted as *infinity*. However, the situation is less straightforward with `bfloat16`, which has the same exponent size with `float32`. In this case, a high-weight norm alone should not pose a problem. However, issues may arise when different components in the network with varying scales are added together, introducing imprecision. These divergences might not be entirely surprising, considering that `bfloat16` offers limited precision. For instance, `bfloat16(256.0) + bfloat16(1.0)` does not yield 257 but rather 256. We suspect this precision limitation is the primary challenge in `bfloat16` runs without weight decay. It is worth noting that reducing the LR can mitigate these issues but results in slower training. Nevertheless, it helps in preventing excessive weight growth, as illustrated in Fig. 20.

## 5 CONCLUSIONS

We find it remarkable that a single hyperparameter can exhibit three distinct effects: providing regularization when paired with stochastic noise, enhancing optimization of the training loss, and ensuring stability of low-precision training. Interestingly, previous work has at times misunderstood or conflated these effects. For instance, while AdamW (Loshchilov & Hutter, 2019) was introduced as a *regularization* method, its popularity in the context of LLMs primarily arises from having a

hyperparameter which is easier to tune in practice. In summary, we conclude that weight decay is seldom valuable as an explicit regularizer; instead, its widespread adoption can be attributed to its ability to induce desirable changes in optimization dynamics.

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

# Appendix

## A    EMPIRICAL CONFIRMATION OF THE CONJECTURES

### A.1    COMPARISON WITH RELATED WORKS

| Paper | Loss function | Algorithm | Implicit regularization |
|---|---|---|---|
| Damian et al. (2021) & Li et al. (2021) | Squared loss & CE + label smoothing | Label noise GD | Trace of Hessian |
| Blanc et al. (2020) | Squared loss | Label noise GD | Jacobian norm |
| Li et al. (2020) | Scale-invariant loss | SGD | - |
| Andriushchenko et al. (2023) | Squared loss | SGD with large LR | Jacobian norm |
| Our work | Regularized CE | SGD with large LR | Trace of Hessian |

Table 1: Comparison of our work with closely related works on regression and implicit regularization phenomenon induced by noise in the algorithm.

### A.2    DISCREPANCY WITH OUR EVALUATION

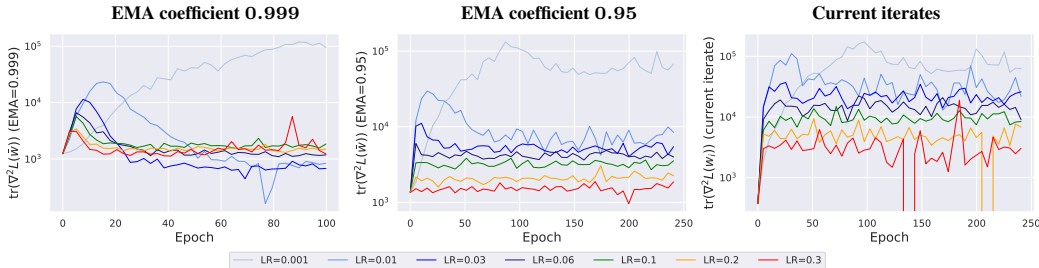

Figure 8: **ResNets-18 trained on CIFAR-10.** We plot $\mathrm{Tr}(\nabla^2)$ at points obtained by weight averaging with different EMA coefficients (0.999 left and 0.95 middle) during the large learning rate phase, together with $\mathrm{Tr}(\nabla^2)$ measured at the current iterates (right). We observe that the ranking between different learning rates supports our conjecture for the current iterates and EMA 0.95 but not EMA 0.999.

In this section, we address some of the concerns raised by the reviewers. The conjecture is primarily motivated by the theoretical works studying implicit regularization of label noise gradient descent (LNGD) on the *squared* loss. From Property **P2**, we know that the noise co-variance has the same shape for both SGD and LNGD. This led us to propose that a similar implicit regularization phenomenon should hold for SGD with large step and weight decay.

Our goal is to show that the iterates of SGD with large step and weight decay $(\mathbf{w}_i)_{i\geq 1}$ stay close to another process $(\tilde{\mathbf{w}}_i)_{i\geq 1}$ which minimizes a regularized objective $L_\lambda + \gamma(\eta)R$ for some regularizing function $R$ and $\gamma(\eta)$ is the strength of regularization. The fine-tuning phase introduced in our framework confirms this idea. Indeed, across the iterations $i$, the iterates $\tilde{\mathbf{w}}_i$'s have comparable loss $L_\lambda$ but the regularizer $R$ decreases with $i$.

We further want to show that the mean of the process $(\mathbf{w}_i)_{i\geq 1}$ converges to a minimizer of the objective $L_\lambda + \gamma(\eta)R$. In this part, we acknowledge a **discrepancy** in our approach. Let $(\mathbf{w}_{i,\eta_1})_{i\geq 1}$ and $(\mathbf{w}_{i,\eta_2})_{i\geq 1}$ be the iterates of SGD with step size $\eta_1$ and $\eta_2$. If our conjecture would hold as formulated, comparing the trace of the Hessian for the iterates of the respective EMAs $(\bar{\mathbf{w}}_{i,\eta_1})_{i\geq 1}$, $(\bar{\mathbf{w}}_{i,\eta_2})_{i\geq 1}$ should show that $R(\bar{\mathbf{w}}_{i,\eta_1}) > R(\bar{\mathbf{w}}_{i,\eta_2})$ for $\eta_1 < \eta_2$ given that we converge to a minimum of $L_\lambda + \gamma(\eta)R$. However, this is not the case as reported in Figure 8 and the behaviour seems to depend on the coefficient of averaging, although the trace across the current iterates $\mathbf{w}_{i,\eta_1}$ and $\mathbf{w}_{i,\eta_2}$ exhibits the predicted trend (see Figure 8, right). It is surprising and in contrast with previous works, to see that $\mathrm{Tr}(\nabla^2)$ for the EMA shows a larger decrease for smaller learning rates. Therefore, we believe that further investigations are needed to clarify this discrepancy. Moreover, our preliminary results reveal that the norm of the Jacobian might be a better candidate for $R$. In the

discussion below we illustrate the close relation between the norm of the Jacobian and the Trace of the Hessian.

**Link between the trace of Hessian and Jacobian norm.**    For any loss function $l$ and a parameterized model $h(\mathbf{w}, x)$ for any input $x$. Consider the empirical loss $L()$ defined by

$$L(\mathbf{w}) = \sum_{i=1}^{N} l\left(y_i, h(\mathbf{w}, x_i)\right).$$

The Hessian of $L$ is

$$\nabla^2 L(\mathbf{w}) = \sum_{i=1}^{N} \left[ \underbrace{\nabla h(x_i; \mathbf{w}) \left[\nabla_h^2 l(h(x_i; \mathbf{w}))\right] \nabla h(x_i; \mathbf{w})^\top}_{G_i(\mathbf{w})} + \underbrace{\sum_{c=1}^{K} [\nabla_h l(h(x_i; \mathbf{w}))]_c \nabla^2 h(x_i; \mathbf{w})}_{E_i(\mathbf{w})} . \right]$$

Many works (Papyan, 2018; Sagun et al., 2017) demonstrated empirically that the $G_i$ is the dominant part of the Hessian decomposition and $\nabla^2 L(\mathbf{w}) \sim \sum_i G_i$.

$$\nabla^2 L(\mathbf{w}) \sim \sum_{i=1}^{N} \left[ \underbrace{\nabla h(x_i; \mathbf{w}) \left[\nabla_h^2 l(h(x_i; \mathbf{w}))\right] \nabla h(x_i; \mathbf{w})^\top}_{G_i(\mathbf{w})} \right]$$

The Jacobian (J) and its norm is defined as

$$\left\|J\right\|_F^2 = \sum_{i=1}^{N} \mathrm{Tr}\left(\nabla h(x_i; \mathbf{w}) \nabla h(x_i; \mathbf{w})^\top\right)$$

In the case of square loss $\left\|.\right\|^2$, $\nabla_h^2 l = I$ where $I$ is the identity matrix. Hence,

$$\mathrm{Tr}\left(\nabla^2 L(\mathbf{w})\right) \sim \left\|J\right\|_F^2.$$

The similarity is an exact equality at an interpolating solution since $\nabla_h l(h(x_i; \mathbf{w})) = 0$. However, in the case of classification, this is no longer true. In particular, since the trace $\nabla_h^2 l$ might be far from the identity and depend on the value of the train loss. We believe this fact to be the **cause of the discrepency** in our evaluation.

**A possible solution.**    As our work is motivated by the theoretical results on regression where the Jacobian norm and the trace of the Hessian of loss function have similar behaviour, we initially conjectured that the implicit regularization term should be $\mathrm{Tr}(\nabla^2)$. **However, it is an oversight on our side as** we did not carefully examine this hypothesis to reveal the correct regularizer $R$. As pointed out before, our experiments with $\mathrm{Tr}(\nabla^2)$ do not provide a complete and consistent picture. Although they reveal a regularized process in close proximity as illustrated by the fine-tuning; the mean shows a discrepancy and therefore should be thoroughly evaluated. Our preliminary experiments on $\left\|J\right\|_F$ show a consistent behavior. Contrary to the $\mathrm{Tr}(\nabla^2)$, the $\left\|J\right\|_F$ of the EMA shows the expected trade-off with the training loss i.e., large learning rate runs have a larger training loss and a smaller norm of Jacobian (Figure 9b). Furthermore, the fine-tuning still shows a regularized process in close proximity (Figure 9a). However, since this observation comes from only one setup (architecture and dataset), we have decided to withdraw the paper and undertake a substantial revision to provide a more comprehensive result.

## B    TRAINING DETAILS

**CIFAR-10/100 experiments.**    We train a VGG network without BatchNorm and preactivation ResNet-18 on CIFAR-10 and ResNet-34 on CIFAR-100 without data augmentations. We use standard SGD *without momentum* for all experiments. We note that $\ell_2$ regularization and weight decay

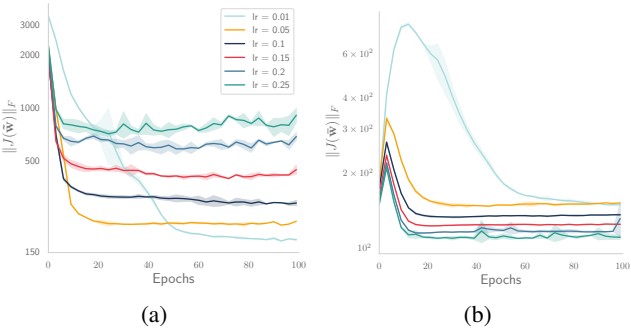

(a)                                          (b)

Figure 9: **Jacobian norm for ResNet-18 on CIFAR-10.** Figure 9a shows a decreasing trend for the fine-tuned iterates and Figure 9b shows the trend for the EMA with the averaging coefficient 0.999.

are exactly the same in this case. We use the standard He initialization (He et al., 2015) for all parameters. To make ResNets scale-invariant, we follow the approach of Li et al. (2020) consisting of fixing the last layer, removing the learnable parameters of the normalization layers and adding a normalization layer in the skip connection. For the experiments in Fig.1, VGG is trained with LR = 0.1 and LR = 0.01 and weight decay parameter is fixed to be either $\lambda_{WD} = 0.0$ or $\lambda_{WD} = 0.008$. The ResNet-18 is trained with LR = 0.08 and LR = 0.001 and $\lambda_{WD} = 0.0$ or $\lambda_{WD} = 0.0125$. The ResNet-34 is trained with LR = 0.15 and LR = 0.001 and weight decay parameter $\lambda_{WD} = 0.0$ or $\lambda_{WD} = 0.01$. The total number of epochs is 1000 in all experiments in Fig.1 and all the LR are decayed at epoch 500 to 0.0001. For the experiments in Fig. 2 we use scale-invariant ResNet-18 and project the SGD iterates on the unitary sphere. We test the following LRs in the large-LR phase $(0.0001, 0.0005, 0.00075, 0.001, 0.002, 0.003, 0.004, 0.005)$ to show different generalization performance. After 100 epochs all the learning rates are decayed to the same value 0.0001. In Fig. 2c and Fig. 2d we fine-tune every 2 epochs for 100 additional epochs with LR=0.0001. For the experiments in Fig. 3 we test three different LRs $(0.1, 0.15, 0, 2)$ and decay all of them to 0.001 after the first 100 epochs. To obtain Fig. 3c and Fig. 3d we fine-tune every 2 epochs for 100 additional epochs with LR=0.001. To compute the trace of the hessian of the model we use the PyHessian library Yao et al. (2020) and consider a subset of CIFAR-10 containing 5000 points. All the experiments are conducted for 5 different random seeds.

**LLM experiments.** We use the `NanoGPT` repository (Karpathy, 2023) for training GPT-2 models (Radford et al., 2019) on OpenWebText (Gokaslan et al., 2019). All training documents are concatenated in a single stream from which a new batch is sampled with replacement on every iteration of training. We train a 124M parameter model known as GPT-2-small for $50\,000$ iterations instead of the default $600\,000$ to make grid searches over the learning rate and weight decay parameters more accessible within an academic budget. We use the context length of 256 in Sec. 4.1 for faster experiments and 1024 in Sec. 4.2 since we observed that a larger context length is crucial to observe loss divergences with moderate learning rates (such as 0.001 for Adam). We train with AdamW (Loshchilov & Hutter, 2019) using batch size 256, default LR 0.0006 (unless mentioned otherwise), $\beta_1 = 0.9$, $\beta_2 = 0.95$, a short 400-iteration LR warmup, and $10\times$ cosine LR decay. For the runs with SGD with momentum, we use the learning rate 0.3 and momentum parameter 0.9 using the same LR schedule as for AdamW. We initialize all parameters with the standard deviation equal to 0.02. We keep all other hyperparameters at their default values as in the `NanoGPT` repository. We perform all experiments on A100 Nvidia GPUs that support fast `bfloat16` training.

## C  WEIGHT DECAY FOR OVERPARAMETRIZED DEEP NETWORKS: ADDITIONAL DETAILS AND FIGURES

### C.1  A GRAPHICAL ILLUSTRATION OF THE FINE-TUNING PHASE

Here, we plot an illustrative graphic in Figure 10 to give an idea of what happens during the fine-tuning phase.

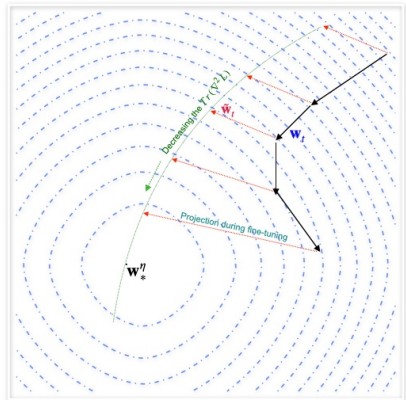

Figure 10: **A graphical illustration of the fine-tuning phase**.

### C.2  SUPPORTING DERIVATIONS

Here we prove that the scale of noise is well approximated by training loss in the case of binary classification instead of classification in the case of multiple classes. The proof follows the lines of Wojtowytsch (2021).

**Proposition 3.** *Assume $\|\mathbf{w}\| \in [a, b]$, for any $x \in \mathcal{D}$, $\|\nabla h(\mathbf{w}, x)\| \in [m, M]$ holds. For $n$ sufficiently large, there exists constants $c_1, c_2$ such that*

$$c_1 \mathcal{L}(\mathbf{w}) \leq \mathbb{E}\left[\|g(\mathbf{w})\|^2\right] \leq c_2 \mathcal{L}(\mathbf{w})$$

*Proof.* The noise in the case when the gradient is computed at $(x_i, y_i)$ is

$$g(\mathbf{w}) = \ell'(y_i, h(\mathbf{w}, x_i))\nabla h(\mathbf{w}, x_i) - \frac{1}{n}\sum_i \nabla \ell'(y_i, h(\mathbf{w}, x_i))\nabla h(\mathbf{w}, x_i),$$

Taking the expectation over uniform sampling over $i$, we have,

$$\mathbb{E}\|g\|^2 = \frac{1}{n}\sum_{i=1}^n \left(\ell'(y_i, h(\mathbf{w}, x_i))\right)^2 \|\nabla h(\mathbf{w}, x_i)\|^2 - \frac{1}{n^2}\|\sum_i \nabla \ell'(y_i, h(\mathbf{w}, x_i))\nabla h(\mathbf{w}, x_i)\|^2 \tag{6}$$

**Upper bound**: Using the self-bounding property of the binary cross entropy, i.e., $(\ell'^2) \leq l$ and $\|\nabla h(\mathbf{w}, x)\|^2 \leq M^2$.

$$\mathbb{E}\|g\|^2 \leq M^2 \frac{1}{n}\sum_{i=1}^n \ell(y_i, h(\mathbf{w}, x_i)) = M^2 \mathcal{L}(\mathbf{w}).$$

**Lower bound**: Again since the iterates are bound, we can assume there exists a constant $c$ such that $(\ell'^2) \geq cl$. as the second term in 6 is decreasing with $O(n^{-2})$, we can assume that the first term is dominating and relevant and can lower bound the first term as,

$$\mathbb{E}\|g\|^2 \geq cm^2 \frac{1}{n}\sum_{i=1}^n \ell(y_i, h(\mathbf{w}, x_i)) = cm^2 \mathcal{L}(\mathbf{w}).$$

This proves the proposition. □

### C.3 Comparison with the related works

Our focus is on an empirical illustration of the implicit regularization phenomenon, hence we refrain from attempting to prove this general conjecture, which we believe is a challenging task. The existing theoretical works Blanc et al. (2020); Li et al. (2021); Damian et al. (2021) present two major weaknesses; they are essentially limiting analysis and as such fail at capturing the *entire optimization trajectory* and they primarily target regression tasks. The powerful mathematical framework for scale-invariant networks developed by Li & Arora (2019); Li et al. (2020) allows them to study in detail the benefits of normalization and its interplay with weight decay. By means of this framework, they state a fast equilibrium conjecture, which gives qualitative guarantees for the speed of convergence of the stochastic process to the stationary distribution in function space. They disentangle the evolution of the norm and the direction of the parameters and show how the evolution of the direction only depends on the intrinsic LR $\lambda_i = \eta\lambda$. However, a qualitative description of the stationary distribution, its dependence on this intrinsic LR and the relationship with generalization is missing (Li et al., 2020, Figure 3(d)). We attempt to fill this gap by providing a qualitative depiction of the stationary distribution and its dependence on the intrinsic LR shading some light towards understanding the relationship with generalization. The work of Kodryan et al. (2022) reports a similar observation, where the best test loss is achieved at a LR where the loss neither converges nor diverges but does not provide any explanation.

### C.4 Additional figures for overparameterized models

In this section, we report additional experimental results related to Section 3 in the main text.

**Training curves for VGG and ResNets.** In Fig. 11 we report the train cross entropy for VGG and ResNet18 on CIFAR-10 and ResNet34 trained on CIFAR-100. We can observe how when weight decay is used in combination with large LR, the train cross entropy stabilizes at some approximately constant level. In Fig. 12 we report the train cross entropy for scale-invariant ResNet on the sphere in Fig. 12b and for standard ResNet trained with weight decay and different large LRs in Fig. 12a. In both cases we can observe different levels of stabilization for the cross entropy depending on the LR deployed in the large-LR phase.

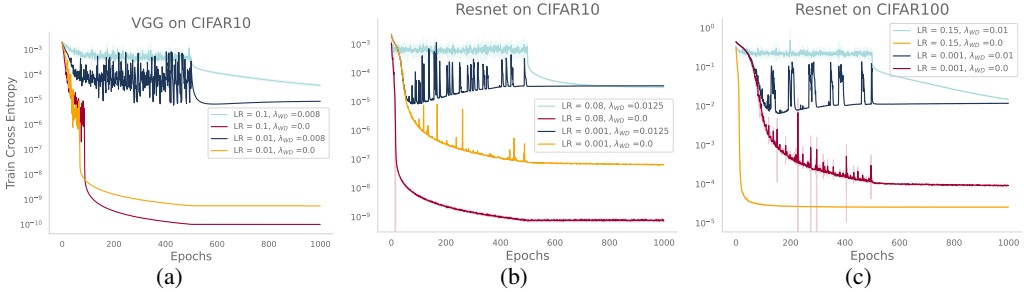

Figure 11: **Training with and w/o weight decay.** We report the train cross entropy for VGG (11a) and ResNet (11b, 11c) trained on CIFAR-10/100 with and without weight decay and with small and large learning rates. After the first 500 epochs the learning rate is decayed to $\eta = 10^{-4}$ for all the curves.

**Connection between SGD covariance and Hessian.** Much of the literature related to implicit bias relies on the assumption that the covariance of the noise of SGD is strictly related to the hessian of the loss function as discussed in Sec 3. Denoting the Hessian $\mathrm{H}(\mathbf{w}) := \nabla^2 \mathcal{L}(\mathbf{w})$ we can write it as the so-called Gauss-Newton decomposition (Sagun et al., 2017; Papyan, 2018) $\mathrm{H}(\mathbf{w}) = \mathrm{G}(\mathbf{w}) + \mathrm{E}(\mathbf{w})$. To measure the cosine similarity (CS) between $\mathrm{w}(\mathbf{w})$ and the covariance $\Sigma_t$ we compute

$$CS = \mathbb{E}\left[\cos\left(\mathrm{H}(\mathbf{w})v, \Sigma_t v\right)\right]$$

where $v$ is sampled from the Gaussian distribution in $\mathbb{R}^p$ and $\cos(u, v) = \langle u, v\rangle / \|u\|\|v\|$. The results are reported in Fig. 13

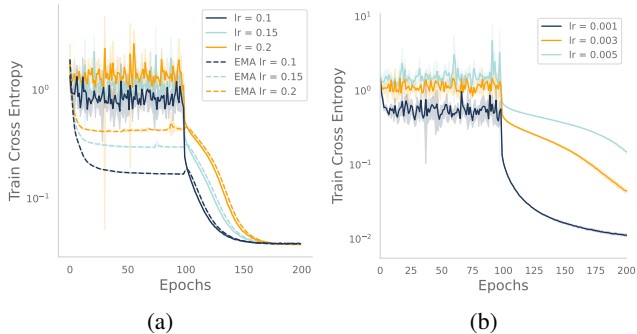

(a)             (b)

Figure 12: **Cross-entropy of standard and scale-invariant ResNets** we train on CIFAR-10 with three different large LR for the first 100 epochs and decay it to $\eta = 10^{-3}$ for the standard ResNets with $\lambda_{WD} = 0.015$ Figure 12a and to $\eta = 10^{-3}$ for the scale-invariant ones 12b.

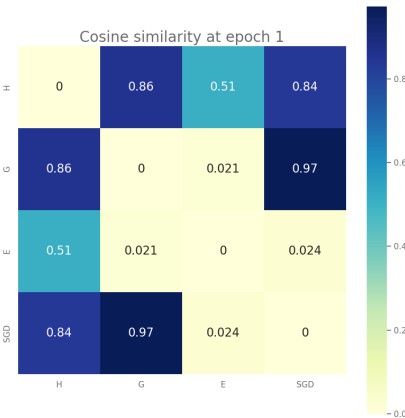

Figure 13: **Cosine similarity between hessian and Noise covariance:** we compute the cosine similarity between the hessian and the covariance of the SGD noise for a scale-invariant ResNet after one epoch with large lr $\eta = 0.005$. The results show how the two matrices are correlated and in particular how the SGD noise covariance is highly correlated with $\mathbf{G}(\mathbf{w})$.

# D    WEIGHT DECAY FOR LARGE LANGUAGE MODELS: ADDITIONAL FIGURES AND DETAILS

## D.1    BIAS-VARIANCE TRADEOFF FOR LLMS AND EFFECTIVE LEARNING RATE

**Effective learning rate induced by weight decay.** Looking at the trend of the norms, we see that different trajectories with different regularizations lead to solutions with different norms. Hence we have a minimizer for the loss at each norm level, hence, the relevant quantity might be the evolution of direction. Here to approximate Adam and to motivate an adaptive learning algorithm, we use the sign stochastic gradient.

$$
\begin{aligned}
\mathbf{w}_{t+1} &= \mathbf{w}_t - \eta_t \lambda_t \mathbf{w}_t - \eta_t \operatorname{sign}(\nabla \ell_t(\mathbf{w}_t)), \\
&= (1 - \eta_t \lambda_t)\, \mathbf{w}_t - \eta_t \operatorname{sign}(\nabla \ell_t(\mathbf{w}_t)), \\
&= (1 - \eta_t \lambda_t)\, \|\mathbf{w}_t\| \left[ \frac{\mathbf{w}_t}{\|\mathbf{w}_t\|} - \frac{\eta_t}{(1 - \eta_t \lambda_t)\, \|\mathbf{w}_t\|} \cdot \operatorname{sign}(\nabla \ell_t(\mathbf{w}_t)) \right]
\end{aligned}
$$

Define $\tilde{\mathbf{w}} := \mathbf{w}/\|\mathbf{w}\|$, using this notation,

$$\tilde{\mathbf{w}}_{t+1} \propto \left[ \tilde{\mathbf{w}}_t - \frac{\eta_t}{(1 - \eta_t \lambda_t) \|\mathbf{w}_t\|} \cdot \mathrm{sign}(\nabla \ell_t(\mathbf{w}_t)) \right]$$

When $\mathrm{sign}(\nabla \ell_t(\mathbf{w}_t))$ is solely determined by the direction $\tilde{\mathbf{w}}_t$, then only the evolution of the direction matters. This holds

- when the norm is constant across iterations, i.e., $\|\mathbf{w}_t\| = c$.
- or the function $\ell$ is scale-invariant or homogeneous, since the following holds

$$\mathrm{sign}(\nabla \ell_t(\mathbf{w}_t)) = \mathrm{sign}(\nabla \ell_t(\tilde{\mathbf{w}}_t)).$$

Looking at the trend of the gradient norm and the norm of the parameters from 17, we see an inverse relationship, i.e., the norm of the gradient is higher when the norm is lower. This is reminiscent of scale invariant networks where $\nabla \ell(\alpha x) = \frac{1}{\alpha} \nabla \ell(\alpha x)$, for any $\alpha \neq 0$.

If the evolution of the direction is the only thing that matters, then it is updated with a time-dependent effective learning rate $\frac{\eta_t}{(1-\eta_t \lambda_t)\|\mathbf{w}_t\|}$. Figure 6 shows the evolution of this effective LR for various runs with and without weight decay. Through the lens of this learning rate, we study the impact of weight decay on convergence of the training loss.

**Understanding experimental observations from the lens of bias-variance tradeoff.** We hypothesize that the effective dynamics induced by weight decay is equivalent to dynamics without any weight decay and a higher learning rate.

- **Constant learning rate.** With a constant learning rate, you will converge to a loss that is eventually dominated by variance and is of $O(\eta)$. Hence as weight decay is equivalent to a higher learning rate, the final loss which will be of $O(\eta_{\mathrm{eff}})$, hence it will be higher and it is confirmed by the Fig. 5.
- **Decaying learning rates.** However with decaying learning rates the picture changes quickly, when the learning rate is large the convergence is still dominated by the variance term, hence the loss in the case of weight decay is higher due to its higher effective learning rate. However, in the small learning rate phase, the bias contraction dominates the convergence, and weight decay quickly catches up and does better again due to its relatively higher effective learning rate and better bias contraction. This explains the trend in the plot on the left of Fig. 5, both towards the end of the training and in the fine-tuning.
- **Testing this hypothesis.** Let's take a run without decay but the cosine schedule is a bit higher. It matches the convergence with weight decay in the final phase indicating that similar artifacts of weight decay can be reproduced using a higher learning rate instead, see Fig. 19. This lends strength to our hypothesis.

## D.2 Additional figures for the LLM experiments

We present the following additional figures related to the LLM experiments.

We show the results for models trained *weight decay on LayerNorm weights* in in Fig. 14. We see that penalizing all parameters in weight decay (i.e., including the LayerNorm parameters) leads to the same effect for smaller $\lambda_{WD}$ (like $0.1$) but underperforms on larger $\lambda_{WD}$ (like $0.3$). Note that when WD is applied on all weights, this changes the optimal value of the objective. In Fig. 15, we train models with $\ell_2$ regularization instead of decoupled weight decay as in AdamW (Loshchilov & Hutter, 2019). We observe that $\ell_2$ regularization instead of weight decay leads to the same effect as decoupled weight decay (Loshchilov & Hutter, 2019). We train models using SGD with momentum and show the results in Fig. 16. We see that weight decay leads to a similar improvement in training loss for SGD with momentum as well. We show multiple metrics in Fig. 17 for the models shown in Fig. 5: gradient variance, gradient norm, and weight norm plots that complement Fig. 6 in the main part. In Fig. 18, we show results of weight averaging that suggests the suboptimality gap between runs with different $\lambda$ is much smaller than what the loss at $w_t$ suggests. However, weight averaging is still less effective than fine-tuning with a tiny LR as in Fig. 5. We show the results of training with longer LR schedules in Fig. 19. We see that slightly larger length of the cosine LR decay leads to a similar effect as weight decay, supporting the effective learning rate view on the role of weight

decay. Note that this experiment is similar to Fig. A1 in Hoffmann et al. (2022). Finally, in Fig. 20, we show results of models trained context length 1024. We see that the training loss over iterations for models trained with a range of LR and WD (all are `bfloat16`). All runs with LR smaller than 0.001 successfully converge but the final training loss is higher than for LR 0.001. In addition, we observe that lower learning rates prevent the weights from growing too much.

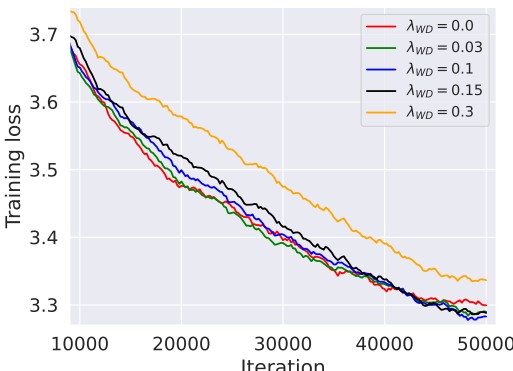

Figure 14: **GPT-2-124M on OpenWebText with weight decay on LayerNorm weights.** Penalizing all parameters in weight decay (i.e., including the LayerNorm parameters) leads to the same effect for smaller $\lambda_{WD}$ (like 0.1) but underperforms on larger $\lambda_{WD}$ (like 0.3). Note that when WD is applied on all weights, this changes the optimal value of the objective.

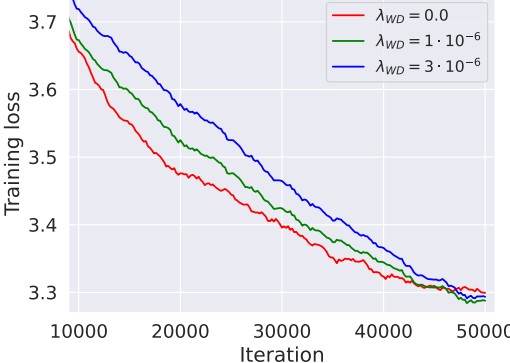

Figure 15: **GPT-2-124M on OpenWebText with $\ell_2$ regularization.** We observe that $\ell_2$ regularization instead of weight decay leads to the same effect as decoupled weight decay (Loshchilov & Hutter, 2019).

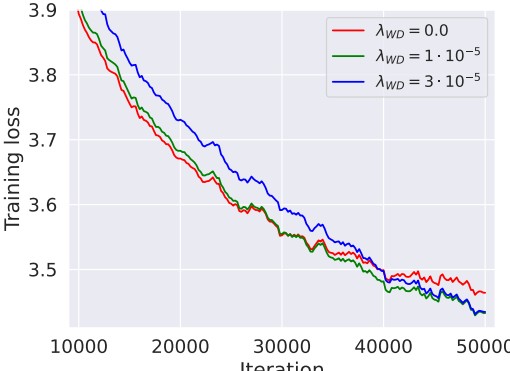

Figure 16: **GPT-2-124M on OpenWebText trained with SGD with momentum.** Weight decay leads to a similar improvement in training loss for *SGD with momentum* as well (all other experiments are done with AdamW).

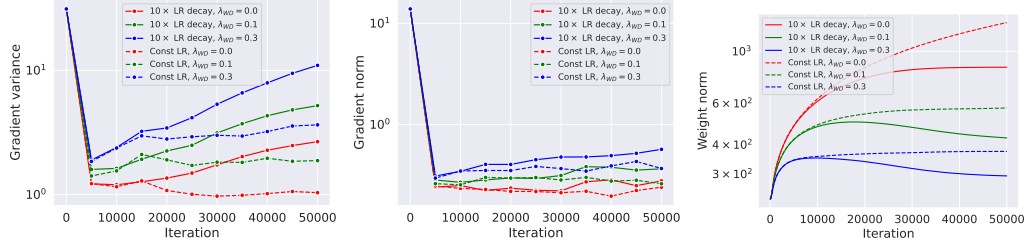

Figure 17: **Multiple metrics for GPT-2-124M on OpenWebText.** Gradient variance, gradient norm, and weight norm plots that complement Fig. 6 in the main part.

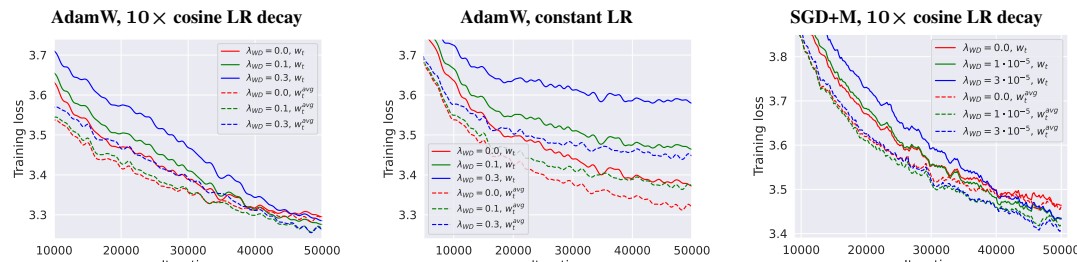

Figure 18: **Weight averaging for GPT-2-124M on OpenWebText.** Weight averaging ($w_t^{avg}$) shows that the suboptimality gap between runs with different $\lambda$ is much smaller than what the loss at $w_t$ suggests. However, weight averaging is still less effective than fine-tuning with a tiny LR as in Fig. 5.

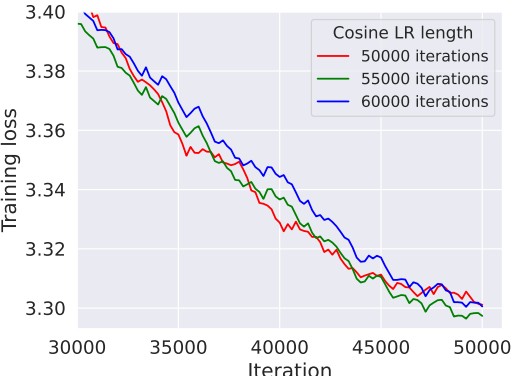

Figure 19: **GPT-2-124M on OpenWebText trained with different LR schedules.** A slightly larger length of the cosine LR decay leads to a similar effect as weight decay, supporting the effective learning rate view on the role of weight decay. Note that this experiment is similar to Fig. A1 in Hoffmann et al. (2022).

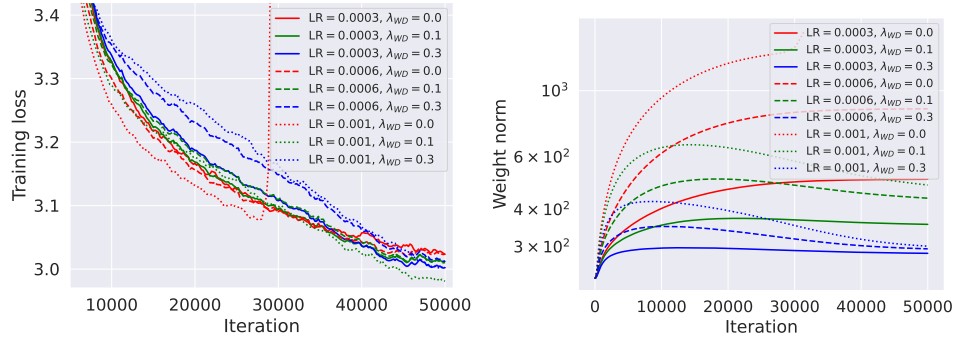

Figure 20: **GPT-2-124M on OpenWebText with context length 1024.** (*Left*) The training loss over iterations for models trained with a range of LR and WD (all are `bfloat16`). All runs with LR smaller than 0.001 successfully converge but the final training loss is higher than for LR 0.001. (*Right*) Weight norms for LR in 0.0003, 0.0006, 0.001 for $\lambda_{WD} = 0.1$ which does not diverge. Lower learning rates prevent the weights from growing too much.

