# OpenReview forum: "Why Do We Need Weight Decay in Modern Deep Learning?"
_ICLR.cc/2024/Conference — ICLR 2024 Conference Withdrawn Submission_

### Official Review · Reviewer_aeLC · 2023-10-16

**Soundness:** 3 good
**Presentation:** 3 good
**Contribution:** 2 fair
**Rating:** 5
**Confidence:** 4

**Summary:**

The paper studies the user of weight decay (WD) in deep learning. There are three main contributions of the paper:

1. The authors state that WD results in implicit regularization by controlling the learning rate. They provide two conjectures which build on empirical observations they have made.
2. The authors replicate findings from (Hoffman el al. 2022), showing that WD leads to lower final loss in the final stages of training LLMs. The authors argue that WD will modulate the effective LR.
3. The authors observe that for a small GPT style model, WD allows better stability when training in bf16. The authors hypothesize that WD alleviates precision issues for bf16.

**Strengths:**

WD is very common, so demonstrating something novel about it would be very impactful.
The paper is rather well written.

**Weaknesses:**

1. The arguments in finding (1) are very handwavy, and they are rather similar to observations done in earlier papers. It would be better if the authors could show proofs and theorems instead of conjectures.
2. Finding (2) from the paper is not novel. The observation that WD will increase the effective LR has been made in (Van Laarhoven, 2017). Furthermore, as stated in the text, the observations on WD for LLMs have been made by (Hoffman el al. 2022)
3. Finding (3) the paper is interesting, but the authors do not really explain why WD allows for better stability in bf16 training. The paper states that “We suspect this precision limitation is the primary challenge in bfloat16 runs without weight decay”. This must be true per definition if fp32 works but bf16 does not, so it’s not a very insightful statement.

**Questions:**

Can you verify your hypothesis for bf16 training?

---

### Official Review · Reviewer_8iM1 · 2023-10-25

**Soundness:** 2 fair
**Presentation:** 2 fair
**Contribution:** 1 poor
**Rating:** 3
**Confidence:** 3

**Summary:**

This paper studies why we need weight decay in modern neural networks. The paper claims that, different from the regularization role in conventional machine learning, weight decay has novel roles and various roles for over-parameterized neural networks and under-parameterized neural networks. For over-parameterized neural networks (CNNs), weight decay modifies the optimization dynamics and enhancing the ever-present implicit regularization of the SGD noise. For under-parameterized neural networks (LLMs trained with nearly one-pass SGD), weight decay balances the bias-variance tradeoff in stochastic optimization leading to lower training loss. Preventing sudden loss divergences for mixed-precision training of LLMs is also reported for weight decay.

**Strengths:**

-	This work focuses on an important topic: the role of weight decay in modern neural networks. Novel insights and observations about weight decay are presented.
-	The empirical results ranging from CNNs to Large Language Models are appreciated. Particularly, the empirical analysis of weight decay on Large Language Models have rarely touched by previous studies.
-	The theoretical results are helpful for understanding the empirical results.

**Weaknesses:**

-	This work divided modern neural networks into two classes: over-parameterized neural networks (using ResNet and VGG) and under-parameterized neural networks (using Large Language Models trained with nearly one-pass SGD). The classification of modern neural networks is very confusing and may be the most significant weakness in this work. I agree that weight decay may have different roles in the two classes of neural networks according to the empirical results. But does over-parameterization and under-parameterization matter here? What is exactly over-parameterization and under-parameterization in this paper. Could it be because ResNet and VGG are CNNs, while Large Language Models are Transformers? Could it be because we train CNNs for many epochs, while we train LLM for only one epoch? Could it because the ratio of the training data size to the number of model parameters?
-	Can weight decay prevent sudden loss divergences for training of neural networks beyond mixed-precision training of LLMs? I believe this does not only happen to mixed-precision training or LLMs. I once observed similar phenomenon for CNNs. As this is a main contribution of this work, it is necessary to further explore this topic.
-	The contributions and conclusions in this work are not informative and quantitative enough. The qualitative conclusions are helpful but not as significant as quantitative ones. For example, what is the quantitative difference between the so-call over-parameterized neural networks and under-parameterized neural networks? The theoretical contributions are also weak due to lack of quantitative analysis and conclusions.
-	A small point. While I think this work did a relatively complete literature review to my knowledge, a recent relevant work on weight decay (https://arxiv.org/abs/2011.11152) is missing in Sec. 2.

**Questions:**

Please see the questions associated to the weaknesses above.

---

### Official Review · Reviewer_Crhm · 2023-10-26

**Soundness:** 2 fair
**Presentation:** 2 fair
**Contribution:** 2 fair
**Rating:** 3
**Confidence:** 3

**Summary:**

This paper studies the effect of weight decay in modern deep learning tasks, ranging from computer vision to language modeling.

**Strengths:**

- The literature review seems complete.

**Weaknesses:**

- Global impression: I think this paper gathers a lot of information and concludes many different things in a succession of paragraphs. It is quite hard to follow the reasoning of the authors as the paper is too dense (or at least should be reorganized).
- As a consequence of the previous point, it is difficult, after reading the paper a few times, to have a clear idea of what are the conclusions of the paper.
- I think vision experiments are a bit too light to conclude, for all modern deep learning, from them. I think a vision transformer, or more complicated dataset (mini Imagenet?) would have strengthened this part of the paper.
- In the end, it is hard to grasp what influences experimental results: architecture, optimizer, dataset? It is not very clear to me after reading the paper.
- It is not very clear to me where the two conjectures come from? I think it is from observations, but is it enough to observe that the loss and the trace of the hessian decrease together (fig 3) to conjecture equation 4 and 5? Did the authors try to observe other important quantities to decorrelate their influence with the trace of the Hessian?
- I don't think the authors give a reason on why weight decay may stabilize the training in bf16? Can they give an intuition?
- I don't think the assumptions of Proposition 3 are reasonable: bounded weights and gradients are strong assumptions. Did the authors try to relax them to let's say, subgaussian? Also, m and M should depend on the dimension of the gradient of h, they are not constant?
- I was surprised by the result on page 8: the loss scaling as $O(\eta)$? This is a very loose bound?

**Questions:**

- I think $\sigma_\eta$ was nowhere defined? (I may have missed it).
- What does "mixing in the function space" mean? When first reading this paragraph, I had the feeling that it was written there just to impress the reviewer/reader with mathematical terms, that are not well defined in the paper.
- I understand the authors may be on a reduced compute budget, but why setting the context length of the GPT2 model to 256? Why not set it to the standard 1024 and reduce the batch size? Due to stability reasons?
- Do the authors clip gradient values in the GPT2 experiments as it seems to be standardly done?
- Page 8: what does "homogeneity of the training loss" mean?
- Page 9: I think there is a typo "with only 7 bits for the fraction instead of 23" ?

---

### Official Review · Reviewer_hANA · 2023-11-02

**Soundness:** 2 fair
**Presentation:** 4 excellent
**Contribution:** 3 good
**Rating:** 5
**Confidence:** 4

**Summary:**

This paper investigates the effect of weight decay in modern deep learning, such as overparameterized models and large language models. The main finding is that, on over-parameterized models, weight decay regularizes the trace of Hessian implicitly, while on LLMs, weight decay mainly benefits the optimization dynamics instead of regularizing the model. Finally, the paper finds the reason for why LLM training has loss spikes from the perspective of the low precision of bfloat16.

**Strengths:**

The paper is well-written and quite interesting to read. It proposes to investigate the important question on the effect of weight decay.

The idea that weight decay helps regularize the trace of Hessian is novel as far as I know.

The section about bfloat16 training of LLMs looks quite interesting.

**Weaknesses:**

Some conclusions in this paper are not quite rigorous, listed as follows.

1. Section 3.3 presents one of the major findings of the paper using just three learning rate values, so it is hard to say whether the conclusion is generalizable to different learning rate settings with a larger range.

2. Section 4 draws a conclusion that weight decay in LLMs is not regularization based on the qualitative result in Fig. 4. This does not look quite rigours, as it is not clear what is the meaning of not regularization and how to tell weight decay is regularization or not based on a quantitative measurement.

3. It is not clear to me how the bfloat16 issue is directly related to weight decay. In other words, how weight decay helps to address the precision issue is not quite well explained.

Overall, I think the paper is quite promising but some important issues should be fixed. I am happy to increase my scores if my concerns and questions can be addressed.

**Questions:**

1. The projected SGD in scale-invariant neural networks is investigated in existing research [1,2,3]. Could the authors provide a discussion on these related research?

2. Third line of Fig. 2's caption should be Fig 2b instead of Fig 2d.

3. I don't quite get the meaning *The model with the largest LR exhibits the smallest $Tr(\Delta)$*, which is contradictory to the result of Fig. 3.

4. I understand that the trace of the Hessian is hard to optimize for large models, is it possible to show the regularization effect for smaller models like two-layer MLPs?

[1] Huang, Lei, et al. "Projection based weight normalization: Efficient method for optimization on oblique manifold in DNNs." Pattern Recognition 105 (2020): 107317.
[2] Liu, Ziquan, et al. "Weight Rescaling: Effective and Robust Regularization for Deep Neural Networks with Batch Normalization." arXiv preprint arXiv:2102.03497 (2021).
[3] Wan, Ruosi, et al. "Spherical motion dynamics: Learning dynamics of normalized neural network using sgd and weight decay." Advances in Neural Information Processing Systems 34 (2021): 6380-6391.

---

### Official Review · Reviewer_i8U2 · 2023-11-04

**Soundness:** 2 fair
**Presentation:** 2 fair
**Contribution:** 2 fair
**Rating:** 5
**Confidence:** 3

**Summary:**

The study explores the necessity of weight decay in deep learning, specifically addressing its relevance and effects. Key contributions of the research include:
1. Weight decay is shown to enhance the implicit regularization of SGD noise in overparameterized networks by maintaining the learning trajectory close to that of a process with a regularized Hessian trace.
2. In LLMs trained with nearly one-pass SGD, weight decay does not act as a significant regularizer but adjusts the effective learning rate and improves the bias-variance tradeoff, reducing training loss.
3. Weight decay also serves to prevent abrupt loss divergences during bfloat16 mixed-precision training, critical for training LLMs at scale.

**Strengths:**

A great deal of research has been done on weight decay.
The authors analyze the role of weight decay in models such as overparameterization and LLM in the context of a large number of existing studies.

**Weaknesses:**

The novelty of this paper is very difficult to understand.
The reason for this is that so many other papers are cited in the analysis section of this paper that it is difficult to tell the difference between them.
The discussion of whether the results are similar or different in existing studies is mixed, so it is not clear whether what is shown in this paper is really something novel.

Also, the third analysis regarding bfloat16 seems to be not so novel.
The paper states the following,

>We suspect that LLM practitioners may be aware of this phenomenon
>qualitatively, but we could not find any systematic reference addressing it.

which does not seem to reveal anything new beyond simply a question of bfloat precision.
 It seems to me that the effect of the weight decay in suppressing large values is simply to suppress overflow.
Is there something more to it than that.
What more is there to say?

**Questions:**

A better way to claim novelty from other papers would be to make a table showing how what is shown in this paper differs from the papers cited in the analysis part.
Would it be possible to make such a table that shows at a glance the differences from existing studies?

---

### Author Response · Authors · 2023-11-23
**Author clarifications**

We thank the reviewers for their constructive suggestions and for identifying some inconsistencies in our paper. We have decided to **withdraw our submission** to further investigate the behavior of the implicit regularization term $\nabla^2 \mathcal{L}(\mathbf w)$ in the conjectures.

Our conjecture is primarily inspired by the **existent theoretical works** studying implicit regularization of label noise gradient descent (LNGD) on the $\textit{squared}$ loss. Nevertheless, during the rebuttal period, we discovered that the behaviour is different. If our conjecture would hold as formulated, the averaged iterates should have a smaller value of the regularizer for larger values of the learning rate (note that this need not happen for the finetuned iterates).  More specifically, although $Tr(\nabla^2)$ is effectively minimized *for the finetuned iterates* as reported in Figure 3 (c) , $Tr(\nabla^2)$ for the EMA does not necessarily have the correct trend with respect to $\eta$ as posited in our conjecture (see Figure 8 in appendix A.2). Furthermore, the trend seems to depend on the coefficient used in the EMA. We can obtain the expected trend for EMA coefficient $0.95$ but not $0.999$. This made us consider the possibility that $Tr(\nabla^2)$ might not be the correct quantity to explain the improvement in generalization when WD is used in combination with large LRs. This intuition is supported by the observation that $Tr(\nabla^2)$ depends on the value of the training loss, potentially clarifying the apparent disparity between the experiments and the conjecture. Attempting to find a quantity that reveals the implicit regularization effect without being directly affected by the value of the training loss, we conducted some preliminary experiments on the Jacobian norm. This quantity closely relates to $Tr(\nabla^2)$ without carrying the explicit dependence on the value of the loss as we elaborate in Appendix A.2. The preliminary experiments show the correct trend of the EMA for standard ResNet-18 on CIFAR-10. We provide a longer discussion on this in Appendix A.2 of the updated version of the paper, together with more details on why we believe both quantities are at play during the large learning rate phase. However, a thorough investigation of the behaviour of the Jacobian norm is needed (i.e., on multiple architectures and datasets) before updating the conjecture.

We would like to briefly clarify other points about our submission:
- **Overparameterization vs. underparameterization.** We agree that this is not the best way to present this distinction and we should have called the one-epoch training of LLMs as the *“underfitting regime”*. What we really meant by (effective) *underparameterization* is that for one-epoch training, the training and validation losses stay very close to each other. We have updated Figure 4 to more clearly illustrate this for LLMs by plotting the generalization gap directly over training iterations for a GPT-2 small and GPT-2 large models.
- **Precise meaning of regularization.** By regularization, we meant any technique that restricts the hypothesis class and reduces the generalization gap. As illustrated in Figure 4, weight decay does not noticeably reduce it for LLMs trained for one epoch (unlike for CIFAR-10 trained for multiple epochs). Thus, we conclude that the role of weight decay is different from reducing the generalization gap. We will use more precise wording to explain what we mean.
- **Novelty of the `bfloat16` observation.** We would like to emphasize that `bfloat16` has the same dynamic range as `float32`, so the issue is not in the overflow (which is an issue, for example, for `float16` whose maximum possible value is $65\\,519$ and larger values are interpreted as `NaN`). We will add a more thorough explanation of this to the paper. Moreover, we will investigate the precise mechanism of how weight decay helps to address the precision issue.
- **Parameters of the LLM experiments.** We used gradient clipping with the default parameter $1.0$ for all experiments so the loss divergence with `bfloat16` is not due to the lack of gradient clipping. We used the context length of 256 primarily to speed up the experiments for Section 4.1 but the overall pattern is the same for the context length 1024.

We thank the reviewers again. These comments will help us to formulate our ideas more precisely and collect more comprehensive experimental evidence for the next version of our paper.